# ADAPTIVE LOSS SCALING FOR MIXED PRECISION TRAINING

## ABSTRACT

Mixed precision training (MPT) is becoming a practical technique to improve the speed and energy efficiency of training deep neural networks by leveraging the fast hardware support for IEEE half-precision floating point that is available in existing GPUs. MPT is typically used in combination with a technique called loss scaling, that works by scaling up the loss value up before the start of back-propagation in order to minimize the impact of numerical underflow on training. Unfortunately, existing methods make this loss scale value a hyperparameter that needs to be tuned per-model, and a single scale cannot be adapted to different layers at different training stages. We introduce a loss scaling-based training method called adaptive loss scaling that makes MPT easier and more practical to use, by removing the need to tune a model-specific loss scale hyperparameter. We achieve this by introducing layer-wise loss scale values which are automatically computed during training to deal with underflow more effectively than existing methods. We present experimental results on a variety of networks and tasks that show our approach can shorten the time to convergence and improve accuracy compared to the existing state-of-the-art MPT and single-precision floating point.

## 1 INTRODUCTION

Training deep neural networks (DNNs) is well-known to be time and energy consuming, motivating the development of new methods and hardware to make training more efficient. One way to improve training efficiency is to use numerical representations that are more hardware-friendly. This is the reason that the IEEE 754 32-bit single-precision floating point format (FP32) is more widely used for training DNNs than the more precise double precision format (FP64), which is commonly used in other areas of high-performance computing. In an effort to further improve hardware efficiency, there has been increasing interest in using data types with even lower precision than FP32 for training (Micikevicius et al., 2018; Kuchaiev et al., 2018; Wang et al., 2018; Kalamkar et al., 2019; Mellempudi et al., 2019; Sakr et al., 2019). Of these, the IEEE half-precision floating-point (FP16) format is already well supported by modern GPU vendors (Choquette et al., 2018). Using FP16 for training can reduce the memory footprint by half compared to FP32 and significantly improve the runtime performance and power efficiency. Nevertheless, numerical issues like overflow, underflow, and rounding errors frequently occur when training in low precision only.

Recent works propose various improvements, of which **mixed precision training (MPT)** (Micikevicius et al., 2018) is the state-of-the-art. Its core idea is to use FP16 for the compute-intensive yet precision-insensitive operations, such as matrix multiplication, for computational efficiency, while using FP32 for the operations that require high precision, such as batch normalization (Ioffe & Szegedy, 2015) and gradient update accumulation. Activations and gradients, which largely contribute to memory consumption, are stored in FP16, while the weights are stored in FP32 for more accurate accumulation of gradient updates. Even though MPT seems promising and has wide support from both hardware and software frameworks, it still suffers from reliability issues, mainly due to the more limited dynamic range of FP16 being unable to adequately cover possible gradient values during training. The most common issue is for small gradients to fall into the underflow gap and become zero, which makes training less effective.

**Loss scaling** (Micikevicius et al., 2018; Kuchaiev et al., 2018; Mellempudi et al., 2019) addresses the range limitation in FP16 by introducing a hyperparameter $\alpha$ to scale the loss value before the

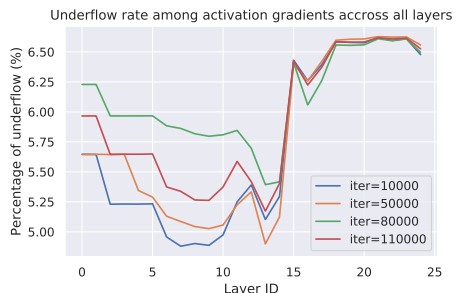
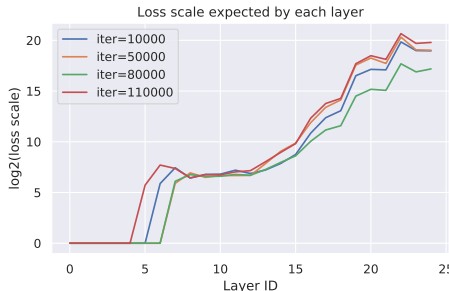

(a) Underflow rate is calculated by counting the absolute gradients below $2^{-24}$, the smallest positive FP16 number.

(b) Expected loss scale of each layer is calculated by 1 over the $(0.01N)$-th smallest absolute gradient, where $N$ is the size of each gradient and $0.01$ is the largest underflow rate permitted.

Figure 1: Statistics of activation gradients collected from training SSD (Liu et al., 2016) by FP32. Data are collected from different training iterations (120k in total). Layer ID are assigned in the topological order of backpropagation computation. Layers with higher ID are closer to the input.

start of the backward pass so that the computed (scaled) gradients can then be properly represented in FP16 without significant underflow. For an appropriate choice of $\alpha$, loss scaling can achieve state of the art results that are competitive with regular FP32 training. Unfortunately, there is no single value of $\alpha$ that will work in arbitrary models, and so it often needs to be tuned per model. Its value must be chosen large enough to prevent underflow issues from affecting training accuracy. However, if $\alpha$ is chosen too large, it could amplify rounding errors caused by **swamping** (Higham, 1993) or even result in overflow. This observed sensitivity to the particular choice of loss scale is also reported by Mellempudi et al. (2019), who find that different values can lead to very different ResNet-50 MPT convergence behavior. Furthermore, the data distribution of gradients can vary both between layers and between iterations (Figure 1), which implies that a single scale is insufficient. For instance, gradients closer to the input require a higher loss scale that may cause overflow or severe rounding errors if the same value were used in layers closer to the output. Including the time spent tuning $\alpha$, the total training time of MPT can even exceed regular FP32 training.

We introduce a loss scaling-based training method called **adaptive loss scaling** that makes MPT easier and more practical to use. We hope that this will help to utilize better existing hardware with support for fast FP16 operations. Our method improves the usability of MPT compared to existing methods by removing the need to tune a model-specific loss scale hyperparameter, while retaining (and in some cases surpassing) the accuracy of regular FP32 training. We achieve this by introducing layer-wise loss scale values which are automatically computed and dynamically updated during training to deal with underflow more effectively than existing methods. Experimental results on several examples show that MPT with adaptive loss scaling can achieve the best model accuracy and the shortest overall training time, especially when training deep models on large datasets.

## 2 BACKGROUND

### 2.1 PRELIMINARY

As mentioned above, MPT (Micikevicius et al., 2018) uses FP16 for storing the activations and gradients and for the most compute-intensive tasks, while FP32 is used only where increased precision is required. FP16 has three fewer exponent bits than FP32, limiting its dynamic range to magnitudes between $u_{min} = 2^{-24}$ and $u_{max} = 65505$. In practice, the gradients often have a larger range than this, resulting in numerical issues when using MPT. In FP16, if the absolute actual value of a gradient $|g|$ is smaller than $u_{min}$, it will become 0; and if it is larger than $u_{max}$, it will be infinite. Also, even if a value is in $[u_{min}, u_{max})$, the closer it comes to either bound, the less accurate its FP16 form is regarding absolute rounding error, e.g., 1024.1 is rounded to 1024. Underflow motivates loss scaling, while the overflow and rounding error are what loss scaling should be careful with.

Figure 2: Comparison between the standard backpropagation algorithm and the loss scaled one. Each layer has a single output in this formulation. `Output` finds the output layer index, `GradW` calculates the gradient update with respect to the weights. `Backprop` calculates the activation gradient given the layer type `OP` and the input gradient.

$\boldsymbol{\delta}_{N+1} \leftarrow$ initial error gradient;
**for** $i \leftarrow$ *layer indices in reversed topological order* **do**
  $\quad j \leftarrow$ `Output`$(i)$;
  $\quad \boldsymbol{W}_i \leftarrow \boldsymbol{W}_i +$ `GradW`$(\boldsymbol{\delta}_j)$;
  $\quad \boldsymbol{\delta}_i \leftarrow$ `Backprop`$($`OP`$(i), \boldsymbol{\delta}_j)$;
**end**

**Algorithm 1:** Standard backpropagation algorithm.

$\boldsymbol{\delta}_{N+1} \leftarrow$ initial error gradient;
$\boldsymbol{\delta}_{N+1} \leftarrow \alpha\boldsymbol{\delta}_{N+1}$;
**for** $i \leftarrow$ *layer indices in reversed topological order* **do**
  $\quad j \leftarrow$ `Output`$(i)$;
  $\quad \boldsymbol{W}_i \leftarrow \boldsymbol{W}_i +$ `GradW`$(\boldsymbol{\delta}_j)$ $/\alpha$;
  $\quad \boldsymbol{\delta}_i \leftarrow$ `Backprop`$($`OP`$(i), \boldsymbol{\delta}_j)$;
**end**

**Algorithm 2:** Standard loss scaling algorithm. $\alpha$ is the loss scale.

Figure 2 shows the basic loss scaling algorithm (Algorithm 2) compared to standard backpropagation without loss scaling (Algorithm 1). Note that they differ only in that in Algorithm 2, the initial error gradients $\delta_{N+1}$ from the output layer are scaled by $\alpha$ before the start of the backward pass, and that the weight gradient update `GradW` is then unscaled by the same $\alpha$ just before the weight update. Recall that $\alpha$ should be chosen large enough to prevent underflow issues from affecting training, while also being small enough to prevent overflow. Even when kept within this range, using a larger value than necessary could introduce more absolute rounding error as aforementioned. Also, since loss scaling amplifies the ratio between the largest and the smallest elements within each gradient, **swamping** (Higham, 1993), the phenomenon that summing small values with larger ones is inaccurate in floating-point, becomes more likely and may hinder training (Wang et al., 2018).

## 2.2 RELATED WORK

Many recent works focus on reducing rounding error to improve the training performance. Wang et al. (2018) devise a chunk-based accumulation mechanism to mitigate the swamping issue. Sakr et al. (2019) improve the solution to the same problem by finding lower precision for accumulation by variance analysis. Alternatively, Hoffer et al. (2018) identify the numerical issues caused by batch normalization and propose to replace it by a more numerically stable and efficient alternative. These methods are orthogonal to loss scaling, and we plan to study the effect of applying them together with adaptive loss scaling as future work.

Loss scaling that aims to improve mixed precision training by reducing the underflow rate in the computed gradients can be traced back to Micikevicius et al. (2018). They originally suggest to choose a constant loss scale either empirically, or using a factor that cannot scale the maximal absolute gradient value to overflow. Kuchaiev et al. (2018) propose two improved versions: one is called *backoff*, which simply makes the loss scale smaller if a numerical error is encountered during training; the other is *logmax*, which models the maximal absolute gradient value across training iterations by log-normal distribution, in order to estimate the proper loss scale value for the *next* iteration. We argue that these new solutions are still not ideal, since backoff is simply trial-and-error and can waste training workload; and logmax is risky to use when we do not have much gradient values to model that log normal distribution, or this assumption cannot apply. Mellempudi et al. (2019) further study the effect of the backoff method for 8-bit floating point.

## 3 ADAPTIVE LOSS SCALING

As a preview, Figure 3 shows a concrete example of our adaptive loss scaling approach in the backward pass of a 3-layer Multi-Layer Perceptron (MLP). After the forward pass has completed, starting from the rightmost node, the gradients $\boldsymbol{\delta}$ are first propagated from loss $\mathcal{L}$ and are then scaled by a scalar $\alpha_4$ before being consumed by the last linear layer. The weight gradients for this layer are then scaled by $1/\alpha_4$ just before the weight update for $\boldsymbol{W}_3$ in order to prevent the particular choice of $\alpha_4$ from affecting the computed gradient magnitudes. It is at this point that our approach begins to differ from the standard loss scaling method. In the standard method, the same $\alpha_4$ would be used

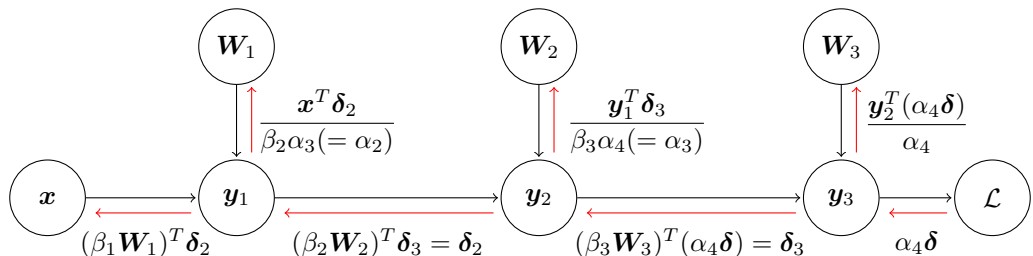

Figure 3: An example of how the adaptive loss scaling method works based on a 3-layer MLP (bias and activation functions are omitted). Black and red arrows represent forward and backward propagation respectively, and terms beside red arrows are gradients. Section 3.1 explains the symbols in this figure, specifically, $\beta_1$, $\beta_2$, and $\beta_3$ denote the loss scale values calculated locally for each layer, and $\alpha_1$, $\alpha_2$, and $\alpha_3$ stand for accumulated scales. $\alpha_4$ is an optional initial scale.

for all layers in the network. However, in our method, each layer $i$ calculates its own local loss scale value $\beta_i$ based on the statistics of its output gradients and weights in the current iteration, in order to minimize underflow in its computed input gradients. This $\beta_i$ is then used to scale weights $W_i$ before computing the scaled input activation gradients for layer $i$. Since the scaling effects from these local loss scales accumulate, when unscaling gradients for updating, we use the product of all previous scale values, indicated by $\alpha_i = \alpha_4 \prod_{j=i+1}^{3} \beta_j$. Thus, our approach attempts to minimize underflow in every layer simultaneously through the use of layer-local loss scales $\beta_i$ which are also computed automatically based on the current layer statistics. Compared to the standard method, this removes the need to perform model-specific hyperparameter tuning and enables layer-wise loss scaling.

### 3.1 Loss Scaled Backpropagation

We use a 2-tuple notation to denote the propagated entity from layer $i$: $\langle \alpha_i, \boldsymbol{\delta}_i \rangle$, in which $\alpha_i$ is the loss scale value for layer $i$ and $\boldsymbol{\delta}_i$ is the gradient that *has been scaled* by $\alpha_i$. To be more specific about this notation, we can take a $N$-layer MLP as an example (see Figure 3 for the notation). In this case, layer $i$ takes in $\langle \alpha_{i+1}, \boldsymbol{\delta}_{i+1} \rangle$, updates its weight by $(\boldsymbol{y}_{i-1}^T \boldsymbol{\delta}_{i+1})/\alpha_{i+1}$, and produces $\langle \alpha_i, \boldsymbol{\delta}_i \rangle$ for the previous layer $i-1$. We will elaborate more on how $\alpha_i$ is calculated in the following sections.

---

**Algorithm 3:** Backpropagation algorithm with adaptive loss scaling, assuming each layer has a single output. Section 3.2.2 shows how multiple outputs work.

---

$\langle \alpha_{N+1}, \boldsymbol{\delta}_{N+1} \rangle \leftarrow$ initial loss scale, and error gradient *scaled* by $\alpha_0$
**for** $i \leftarrow$ *layer ID in a reversed topological order* **do**
$\quad j \leftarrow$ GetLayerOutput($i$)
$\quad W_i \leftarrow W_i +$ GetWeightGradient($\boldsymbol{\delta}_j$) $/\alpha_j$
$\quad \beta_i \leftarrow$ GetLossScale(OP($i$), $\langle \alpha_j, \boldsymbol{\delta}_j \rangle$)
$\quad \langle \alpha_i, \boldsymbol{\delta}_i \rangle \leftarrow \langle \alpha_j \beta_i,$ Backprop(OP($i$), $\beta_i \boldsymbol{\delta}_j) \rangle$
**end**

---

Algorithm 3 shows the pseudocode for adaptive loss scaled backpropagation for the case where each layer has a single output (we describe how to handle the multiple-output case in Section 3.2.2):

1. We start with the error gradients $\delta_{N+1}$ computed from the output loss value for the last layer $N+1$. We may optionally scale this gradient by $\alpha_{N+1}$. Normally we keep it as 1.
2. As visiting each previous layer $i$ in a reversed topological order of the computational graph, we retrieve the tuple $\langle \alpha_j, \boldsymbol{\delta}_j \rangle$ propagated to it from the next downstream layer that represents the scaled loss for layer $i$'s output. We calculate a local loss scale value $\beta_i$, which will be used to scale $\boldsymbol{\delta}_j$ *before* we calculate the activation gradients for the previous layer.
3. We use $\boldsymbol{\delta}_j$ and other cached inputs (omitted) to compute the gradients for $W_i$. However, since these gradients have been scaled, we must unscale them using $\alpha_j$ before performing the weight update.

4. Since $\beta_i$ contributes to the magnitude of the gradient $\boldsymbol{\delta}_i$, we calculate the loss scale value $\boldsymbol{\delta}_i$ to be passed to the next previous layer as $\alpha_j \beta_i$.

## 3.2 Loss Scale Calculation

This section describes how to compute the layer-wise loss scales for the various operation types that are sufficient to support its implementation in general DNNs, which broadly consists of general matrix multiplication (**GEMM**) and **element-wise** operations. GEMM is the basis of linear layers, while the element-wise category covers batch normalization (Ioffe & Szegedy, 2015), activation functions, and math operations such as the element-wise addition commonly used in skip connections (He et al., 2016).

### 3.2.1 GEMM

We start by considering a linear layer with input activations $\boldsymbol{X}$, weights $\boldsymbol{W}$, and output activations $\boldsymbol{Y}$. The GEMM computation for the forward-pass is then given by $\boldsymbol{Y} = \boldsymbol{X}\boldsymbol{W}^T$ (ignoring the bias term without loss of generality). Now consider the backward pass for this same layer, in which the "input" to the layer consists of the tuple $\langle \alpha, \boldsymbol{\delta} \rangle$ received from the next downstream layer, where $\boldsymbol{\delta} = \alpha \frac{\partial L}{\partial \boldsymbol{Y}}$. Note that $\alpha$ represents the total loss scale (i.e., product of all downstream layer-wise scales). As shown earlier, the weight gradients for $\boldsymbol{W}$ are computed by $(\boldsymbol{X}^T \boldsymbol{\delta})/\alpha$.

We assume that both $\boldsymbol{W}$ and $\boldsymbol{\delta}$ can be characterized by two i.i.d. normal random variables w and g, respectively, so that w $\sim \mathcal{N}(\mu_w, \sigma_w^2)$ and g $\sim \mathcal{N}(\mu_g, \sigma_g^2)$. We also assume that their product p = wg is another random variable with a zero-mean normal distribution, i.e., p $\sim \mathcal{N}(0, \sigma_p^2)$. These assumptions are standard in the literature on weight initialization (He et al., 2015; Glorot & Bengio, 2010) and low-precision floating-point training (Sakr et al., 2019). Since w and g are uncorrelated, $\sigma_p^2$ is given by $(\sigma_w^2 + \mu_w^2)(\sigma_g^2 + \mu_g^2)$ based on product distribution rules, and can be computed from the corresponding empirical statistics.

p characterizes the distribution of the intermediate results before the final GEMM reduction happens, so that the output is the sum of $N$ values sampled from p, where $N$ is the number of columns. Intuitively, if the probability that p experiences underflow is reduced, the final result will have less underflow rate as well. Let the upper bound for an underflow positive value be $u$, which can take the minimal subnormal value of the given low-precision data type, e.g., $u = u_{min} = 2^{-24}$ for FP16, then our objective corresponds to reducing $P(|\mathrm{p}| \leq u)$ by scaling w or g.

$$P(\alpha|\mathrm{p}| \leq u) \leq T_{uf} \Leftrightarrow \mathrm{erf}\left(\frac{u}{\alpha \sigma_p \sqrt{2}}\right) \leq T_{uf} \Leftrightarrow \alpha \geq \frac{u}{\sigma_p \sqrt{2} \times \mathrm{erf}^{-1}(T_{uf})} \tag{1}$$

We introduce a new term, $T_{uf}$ that specifies the threshold for the probability of underflow for p in each layer, which can also be interpreted as the upper bound of underflow rate. Suppose either w or g is scaled by $\alpha$ before the multiplication, then $P(\alpha|\mathrm{p}| \leq u) \leq T_{uf}$ becomes the expected outcome of loss scaling. Since p is assumed to be $\mathcal{N}(0, \sigma_p^2)$, it implies that $|\mathrm{p}|$ is a random variable with half-normal distribution, and $P(\alpha|\mathrm{p}| \leq u) = \mathrm{erf}(u/(\alpha \sigma_p \sqrt{2}))$. Therefore, we can deduce the **lower bound** of loss scale for each GEMM-based layer by equation 1 with $T_{uf}$ and $u$.

GetGEMMLossScale($\boldsymbol{W}$, $\alpha \boldsymbol{\delta}$, $u$, $T_{uf}$)
   $\mu_w, \mu_g, \sigma_w, \sigma_g$
     $\leftarrow \mathbb{E}[\boldsymbol{W}], \mathbb{E}[\alpha \boldsymbol{\delta}], \mathrm{Var}[\boldsymbol{W}], \mathrm{Var}[\alpha \boldsymbol{\delta}]$
   $\sigma_p^2 \leftarrow (\sigma_w^2 + \mu_w^2)(\sigma_g^2 + \mu_g^2)$
   **return** $u/(\sigma_p \sqrt{2} \times \mathrm{erf}^{-1}(T_{uf}))$
**Algorithm 4:** Loss scaling for GEMM.

GetBranchLossScale($\{\langle \alpha_i, \boldsymbol{\delta}_i \rangle\}_{i=1}^N$)
   $\{\alpha_i\}_{i=1}^N \leftarrow \mathrm{DescSort}(\{\alpha_i\}_{i=1}^N)$
   **for** $i \leftarrow 1, \ldots, N$ **do**
     **if** $\forall 1 \leq j \leq N$,
     $\alpha_i \max(|\boldsymbol{\delta}_j|) < u_{max}$ **then**
       **return** $\{\langle \alpha_i, (\alpha_i/\alpha_k)\boldsymbol{\delta}_k \rangle\}_{k=1}^N$
   **end**
**Algorithm 5:** Loss scaling for branches.

In practice, we take this lower bound term as the loss scale value (Section 3.2.3 presents the details for the corresponding upper bound to prevent overflow). Algorithm 4 illustrates the steps of the loss scale calculation for a GEMM-based layer. Note that the computation of $\sigma_p$ requires the statistics

of w and g. These in turn require computing the sample mean and variance of $\boldsymbol{W}$ and $\boldsymbol{\delta}$, which is the main source of computational overhead, with around the same computation budget as batch normalization. In our current implementation, these statistics are calculated on GPU and then transferred to CPU to finish the loss scale calculation. There is potential to optimize the data transfer mechanism in the future, and for now we simply reduce the frequency of loss scale update if the overhead is large. For FP16-based mixed precision training, we set $u$ to $2^{-24}$ as required by the FP16 representation range. $T_{uf}$ is set to $1.0 \times 10^{-3}$ in all of our experiments, which corresponds to allowing an underflow rate of 0.1%.

### 3.2.2 ELEMENT-WISE AND BRANCHING OPERATIONS

Element-wise operations can take one input argument (unary), such as activation functions; or two operands (binary), e.g., element-wise multiplication and addition. Batch normalization also falls into this category. Heuristically, we do not update the loss scale for these operations, because normally they will not significantly change the amount of underflow values, and there are no statistical properties that we can directly make use of without introducing much computational overhead.

One particular element-wise operation that requires special treatment is **branching**. It is used mainly in networks that employ skip connections, such as ResNet (He et al., 2016), DenseNet (Huang et al., 2016), etc.; it also appears in object detection models that have multiple outputs, such as SSD (Liu et al., 2016). It copies its single input to multiple branches in the forward pass, and sums all received gradients during backpropagation. The special case that we should treat deliberately is when the gradients to sum have different loss scales. If we were to directly sum these gradients, we would no longer be able to compute the output gradient's loss scale, preventing subsequent layers from restoring the correct gradient magnitude in their weight updates.

Our solution to this issue is to **rescale** input gradients before the summation happens, as shown in Algorithm 5. Suppose we have $N$ input tuples $\{\langle \alpha_i, \boldsymbol{\delta}_i \rangle\}_{i=1}^{N}$, and $\alpha_j$ is the maximum loss scale among them, then the rescaling works by multiplying any gradient other than $\boldsymbol{\delta}_j$ by a factor to produce the gradient that has the same loss scale as $j$, which is calculated by $\alpha_j/\alpha_i$. And if $\alpha_j$ is too large and may cause overflow in other gradients, which is decided by checking the scaled maximum absolute gradient, we will search in a descending order of all scales among inputs until we find one.

### 3.2.3 POST-PROCESSING

A raw loss scale value calculated from Equation 1 should be post-processed by the following rules. Most importantly, the raw loss scale should be rounded down to the nearest **power-of-two** number. Otherwise, due to the nature of floating-point, a scaled gradient cannot always be unscaled by the same scale, i.e., $(\alpha x)/\alpha \neq x$ if $\alpha$ is not a power of two (Muller et al., 2010).

The upper bound of layer-wise loss scale is determined by avoiding overflow. For the GEMM case, it can be simply calculated by choosing the maximal numbers from both operands, multiplying them together, and then taking $u_{max}$ over that multiplication result as the largest possible loss scale, i.e., $u_{max}/(\max(\boldsymbol{W}) \times \max(\boldsymbol{\delta}))$. This is a loose bound since these selected maximal numbers may not be multiplied together when calculating the output activation gradient. In practice, this upper bound is much larger than the lower bound, and we simply choose the lower bound as the loss scale value, and only switch to the upper bound only when the upper bound is smaller than the lower bound. Other operators will not update loss scale (except branching, which has been discussed in Section 3.2.2), we assume they will not cause overflow.

## 4 EXPERIMENT

This section presents various experiments to show the benefit of using adaptive loss scaling over other loss scaling approaches. The models we focus on are basically for computer vision tasks, including image classification and object detection, and their topologies vary, ranging from sequential architecture to skip connection based ones. We implemented our approach using Chainer v6.1.0 (Tokui et al., 2019), a deep learning framework that supports efficient automatic differentiation. For comparison, we choose FP32 training, loss scaling by a fixed value (Micikevicius et al., 2018), and dynamic loss scaling by backoff (Kuchaiev et al., 2018) as baselines.

## 4.1 CIFAR

We first evaluate our method on CIFAR-10/100 (Krizhevsky, 2009) image classification using ResNet models (He et al., 2016) of depth 20, 56, and 110. We explore three different loss scaling options: without loss scaling, fixed loss scaling (Micikevicius et al., 2018) with scales selected from $\{16, 128, 1024, 4096, 8192, 16384\}$, and adaptive loss scaling ($T_{uf}$ is set to $1e^{-3}$). The other training settings are the same as specified in the original paper (He et al., 2016).

Table 1: Test accuracy results for ResNet models trained on CIFAR-10/100, each **averaged** from 4 runs with different random seeds. The best result for each combination of dataset and model is bolded. Fixed loss scaling is tied with ours for the best result on ResNet-20 (C100).

| CIFAR | Depth | FP32 | None | Fixed (best) | Fixed (worst) | Adaptive |
|-------|-------|------|------|--------------|---------------|----------|
| | 20 | 91.25% | 92.16% | **92.24%** | 92.16% | **92.26%** |
| C10 | 56 | 93.03% | 92.79% | **93.29%** | 92.78% | 93.22% |
| | 110 | 93.57% | 93.54% | 93.85% | 93.73% | **93.90%** |
| | 20 | 67.94% | 68.31% | **68.48%** | 68.18% | **68.47%** |
| C100 | 56 | 71.15% | 71.17% | **71.56%** | 71.26% | 71.26% |
| | 110 | 71.14% | 72.05% | 72.46% | 72.34% | **72.66%** |

Results are in Table 1. First of all, ResNet models overfit on CIFAR (train accuracy reaches 100%), such that reducing arithmetic error may decrease its regularization effect and then result in worse test accuracy. That is why adaptive loss scaling is not very beneficial for ResNet-20 and 56. But regarding ResNet-110, since it is much deeper than the others, gradients are harder to propagate and underflow is more harmful, and arithmetic errors hinder training rather than regularizing it. More importantly, in terms of the total training time, to find the best fixed loss scale we should train **6 times to cover all candidates**, while adaptive loss scaling only needs one round.

## 4.2 IMAGE CLASSIFICATION

We also compare loss scaling methods on ILSVRC2012 (Jia Deng et al., 2009). ResNet-18 and 50 (He et al., 2016) are baseline. Based on the training method proposed in the original paper, we set the data type of each layer by the default mixed precision training setting, and change only the loss scaling method. Due to the limitation of resources, we can only select 128 as the fixed loss scale. We also compare with dynamic loss scaling using the backoff strategy (Kuchaiev et al., 2018).

Table 2: Image classification evaluation for different loss scaling methods. Numbers showed here are top-1 test accuracy (%). None means no loss scaling applied.

| Model | FP32 | None | Fixed (128) | Dynamic | Adaptive |
|-------|------|------|-------------|---------|----------|
| ResNet-18 | 69.76 | 71.24 | 71.39 | 71.39 | **71.44** |
| ResNet-50 | 76.15 | 76.07 | 76.02 | 76.12 | **76.22** |

Results are listed in Table 2. In general, adaptive loss scaling performs the best among all MPT and FP32 training results. Note that loss scaling with a fixed arbitrary scale (128) even reduces model test accuracy for ResNet-50 compared with no loss scaling, and on the contrary, there is no hassle in hyperparameter using adaptive loss scaling. Loss scale of each layer in ResNet-18 is listed in Figure 4a. It shows that our calculated loss scale is much smaller than 128, which implies 128 is too large and may cause rounding error. We further compare the **maximum standard deviation** $\sigma_{max}$ of all gradients at different iterations between adaptive and fixed, and we observe that the ratio of $\sigma_{max}$ of fixed over adaptive is around **20 times**, both for ResNet-18/50, which can be the major cause for accuracy drop since that high variance can increase accumulation error (Sakr et al., 2019).

## 4.3 OBJECT DETECTION

We select the Single-Shot Detector model (Liu et al., 2016) SSD512 (VGG-16 backbone, 512 input resolution) as our baseline for the object detection task. The basic training schedule stays the same as the original paper. SSD is a rather challenging model for MPT: its VGG-16 backbone is not

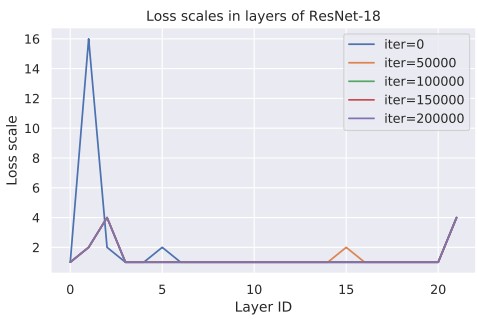
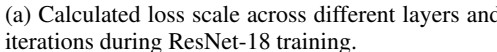
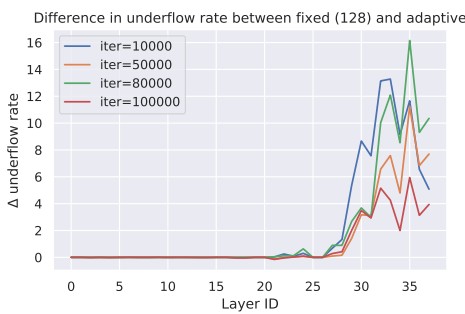

(a) Calculated loss scale across different layers and iterations during ResNet-18 training.

(b) The difference in underflow rate of each layer's activation gradient comparing fixed and adaptive loss scaling.

Figure 4: Examples that show the benefit from using adaptive loss scaling. Note that the underflow rate in Figure 4b is collected by subtracting the percentage of zeros of the FP16 result and the cast-to-FP32 result. The higher $\Delta$ is, the more effective that adaptive loss scaling can mitigate underflow.

interleaved by batch normalization layers, which implies that gradients are not normalized, and their distribution can vary a lot across layers at different depth. It also has a multi-branched topology, in which each branch detects objects at a different scale and passes different values. As seen in Figure 1, SSD512 cannot be properly scaled by a fixed loss scale.

Table 3: Test performance of SSD512 models trained by different loss scaling methods, measured in mAP (%). FP32 denotes the baseline results trained in FP32, and we select the golden value from (Fu et al., 2017) for the case using 32 as the batch size. None means no loss scaling is used, "Fixed (best)" stands for the best performance we can find after trying out several fixed loss scale values (including {8, 128, 1024, 2048}), and "Dynamic" shows the results of dynamic loss scaling.

| Batch | FP32 | None | Fixed (best) | Dynamic | Adaptive |
|-------|------|------|--------------|---------|----------|
| 8 | 78.94 | diverged | 79.11 | 75.04 | **79.24** |
| 32 | 79.50 | diverged | 80.01 | 80.17 | **80.31** |

Table 3 shows the comparison result. We examine two scenarios with different batch sizes yet the same total number of iterations. Without loss scaling training diverges at a very early stage. Loss scaling can make mixed precision training stabler, e.g., the best fixed loss scaling results perform similarly to the FP32 baseline. When the batch size is small, dynamic loss scaling performs poorly since NaN is more frequent and the current dynamic algorithm finds it hard to choose a good scale.

Our adaptive method gives the best result even compared to the FP32 baseline. Figure 4b shows the large amount of underflow rate reduced by using adaptive loss scaling, which can give a hint about why it performs better. Regarding the speed overhead of calculating loss scale, computing the statistics takes around 27% of the overall training time if we update loss scale per iteration. The update frequency of adaptive loss scaling results per 100 iterations, which reduces the overhead to 0.27%. Compared to fixed loss scaling, which performs worse and requires 3 rounds of training to find the best scale, our approach seems appealing for the reduction in total training time it provides.

## 5 CONCLUSION

This paper presents adaptive loss scaling, a method that calculates layer-wise loss scale during run-time, to improve the performance and usability of MPT. Empirically we find it works better than plain MPT, existing loss scaling methods, and even FP32 in some cases, regarding model accuracy and the time taken to converge. Future work includes evaluating adaptive loss scaling on other tasks and models, especially those for Natural Language Processing; and trying to find a tighter upper bound of loss scale for each layer, e.g., based on the variance analysis in (Sakr et al., 2019), such that each layer can be scaled more effectively; extending it to FP8 is also intriguing to try.

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

## A    DETAILED ANALYSIS ON CIFAR RESULTS

Table 1 shows that adaptive loss scaling is beneficial for training ResNet-110, while less advantageous for ResNet-20 and ResNet-56. We hypothesize the reason behind is that underflow causes more numerical problems when the model is deeper. For shallower models, the difference between the oracle gradient values and the underflowing ones is moderate and can even be viewed as a form of regularization. This argument is supported by the fact that the training accuracy of ResNet models on CIFAR can always reach 100%. In this way, even though adaptive loss scaling can improve the accuracy of the computed gradients, this does not necessarily always translate to improved test accuracy.

Table 4: The effect of different fixed loss scales on the test accuracy, which is measured for ResNet-20 and ResNet-56 on CIFAR-10. Numbers on the first row give the fixed loss scales.

| Model | 1 | 16 | 128 | 1024 | 4096 | 8192 | 16384 |
|---|---|---|---|---|---|---|---|
| ResNet-20 | 92.16% | 92.19% | 92.16% | 92.20% | 92.24% | 92.24% | 92.24% |
| ResNet-56 | 92.80% | 93.28% | 92.79% | 93.08% | 93.19% | 93.19% | 93.19% |

We dive deeper into this argument by reviewing Table 4, which shows the test accuracy of the two shallower ResNet models on CIFAR-10. For both models, the test accuracy first increases to a maxima at 16, then there is a sudden drop at 128, and finally it climbs up to a plateau. Our hypothetical interpretation is as follows:

1. Initially the test accuracy is low. Here the underflow rate is expected to be at its highest, and it is the major cause for the low test accuracy.
2. The test accuracy then increases with loss scale, mainly due to the mitigation of underflow by loss scaling. However, as the gradients become more accurate, the regularizing effect from underflow is also reduced and the test accuracy will drop, until the loss scale reaches around 128.
3. If the loss scale continues to increase, the high rounding error and swamping problem caused by large scales will arise. It adds another kind of regularization, which is relatively more harmful than what underflow may cause, and the test accuracy cannot improve much.

Even though this interpretation is hypothetical, this empirical evaluation in Table 4 shows that the relationship between the goodness of a loss scaling scheme and test accuracy is complicated when the model tends to overfit.

## B    EFFECT FROM DIFFERENT LOSS SCALES ON SSD

Here we present how changing fixed loss scales will affect the SSD training result, in order to understand the benefits of both training time and model accuracy from using adaptive loss scaling.

Table 5: Changes in mAP after changing the fixed loss scales. Batch size = 8.

| Loss scale | 1 | 8 | 16 | 128 | 1024 | 2048 |
|---|---|---|---|---|---|---|
| mAP (%) | diverged | 24.62 | diverged | 79.04 | 79.04 | 79.11 |

Table 5 gives all the current empirical results. No loss scaling and fixed loss scaling with scale as 16 are both diverged. 8 almost does not improve during training. {128, 1024, 2048} all give good results compared to other scale candidates.

