# OpenReview forum: "Adaptive Loss Scaling for Mixed Precision Training"
_ICLR.cc/2020/Conference — Reject_

### Official Review · AnonReviewer4 · 2019-10-19
**Official Blind Review #4**

**Rating:** 3

**Review:**

The authors propose an adaptive loss scaling method during the backpropagation stage for the mix precision training to reduce the underflow. Compared with the previous work, which scales the loss by human design, and needs to be consistent in all layers. The authors state that they can decide the scale rate layer by layer automatically to reduce the underflow in a low precision situation.

They calculate the scale rate using the statistic information of the layer weight and gradient. By adaptively scale each layer’s loss and gradient, this method can reduce the underflow rate better than the previous work. Additionally, the authors claim that the computation overhead is not significant, so it is efficient to use rather than searching from a set of fix scale rates.

The experiments present on image classification and objective detection benchmarks. From the result, we can see that the adaptive loss scale reaches a relatively high point on all the tasks.

Pros:

- The method is straight forward and easy to understand. The motivation is good. They get some impressive results on ResNet110 and SSD512 comparing with the fixed scaling method. Besides, they give some analysis of their advantages and disadvantages in different networks, which looks promising to me.

Cons:

- One question in Section 3.2.1, the assumption that w, g, p can be treated as the random variable with Gaussian distribution seems not natural in the training process. Especially p is a zero-mean distribution. The cited paper uses this assumption in a more convincing case, such as the weight initialization task. Notice that He et al., 2015 claim that the product of weight and gradient can be a zero-mean normal distribution is based on the weight is a symmetric distribution around zero, which is not true in neither this paper’s assumption nor the real training situation.

- In the objective detection part, I can not find which dataset the authors use. Though the author state that they follow Liu et al., 2016 ’s work, there are also several tasks in Liu’s paper, and I can not directly match the resulting point with any of those tasks, which makes me hard to confirm the experiment result.

- The experiment setting is unclear. Here are two questions. 1, What is the initial scale at the last layer? It should be manually designed in the experiment, and I think this value may affect the other layer’s scale as well. If the algorithm is robust for this scale, it is better to show some study on that. 2, What update frequency is used in the experiment? The authors say that the overhead can be reduced by reducing the frequency, but they do not clearly show which frequency they use in their experiment, if the frequency does not affect the performance, it is also better to claim or show some study on that.

Minor comments:
- Figures can use a larger font.  Figures 4a and 4b can be aligned better.

**Experience Assessment:**

I have read many papers in this area.

**Review Assessment: Checking Correctness Of Derivations And Theory:**

I assessed the sensibility of the derivations and theory.

**Review Assessment: Checking Correctness Of Experiments:**

I assessed the sensibility of the experiments.

**Review Assessment: Thoroughness In Paper Reading:**

I read the paper thoroughly.

---

> ### Author Response · Authors · 2019-11-06
> **Response for reviewer #4**
>
> Thank you so much for your retailed review.
>
> Q1:
> This is a valuable comment. We do understand that this assumption on distribution may cause some confusion points, and we don’t have enough space to explain them in detail in the paper. Here we elaborate more on this topic.
> First of all, regarding the source of this assumption: the assumption on the product term to be with normal distribution is not just from [1], [2] also establishes their theoretical framework on this assumption and achieves good empirical evaluation results, which provides another source of confidence for using this assumption.
> Next, it is true that a statistical property that holds in the initialization phase may not still hold during the training procedure. But we haven’t found any existing paper that can model tightly how the data distribution may change during training, and we unfortunately don’t have the required mathematical background to achieve that goal either. Therefore, in this paper, we conservatively argue that, if the assumption holds, our calculated loss scale can perfectly achieve the desired underflow rate; and if not, we may exceedingly or not effectively reduce the underflow rate, but overflow is guaranteed not to happen (Section 3.2.3).
> Last but not least, we make some empirical discoveries on this assumption. We run ResNet-18 training on ImageNet for several iterations, and we collect the product term from different layers at some iterations and plot their distribution (Fig 2. in https://github.com/ada-loss/ada-loss/issues/1). In this figure, we present the histogram of actual data, as well as the data sampled from a normal distribution, which is parameterized by the mean and standard deviation of the actual data. It shows that for most layers, the distribution of the actual data looks similar to the modelled normal distribution. Also, as shown in the subtitle of each figure, these actual data are quite close to zero mean. Indeed, there are biases in real-world data from the assumption, but since these biases are moderate (e.g., not a completely different distribution), and our implementation can permit these biases, the usability of our approach won’t be affected much.
>
> Q2:
> Sorry for not explicitly mentioning these details.
>
> Our reference implementation is https://github.com/chainer/chainercv/tree/master/examples/ssd, which uses PASCAL VOC 2007 + 2012 as the training dataset, and PASCAL VOC 2007 for validation. Please refers to that example for more information. You can also confirm details by checking our repo: https://github.com/ada-loss/ada-loss/tree/master/examples/adaptive_loss_scaling/ssd
>
> Q3.1:
>
> That initial scale is always set as 1 for all of our adaptive loss scaling experiments. We mention this term mainly to adapt our adaptive loss scaling formulation to the standard loss scaling approach. It can be completely removed.
>
> Q3.2:
> The update frequency for CIFAR and ImageNet training is per 1000 iterations, and for object detection it is per 100 iterations. Update frequency is, as far as we've empirically confirmed, neutral to accuracy. For example:
> Updating loss scale per iteration gives 80.30 mAP (%) for SSD, and this value is 80.31 for per 100 iterations update.
> For ResNet-18 on ImageNet, updating per 1000 iterations gives 71.44 test accuracy, while for per iteration update the value is 71.48. We will conduct more experiments regarding the frequency.
>
> [1]. He, K., Zhang, X., Ren, S., & Sun, J. (2015). Delving deep into rectifiers: Surpassing human-level performance on imagenet classification. ICCV, 1026–1034.
> [2]. Sakr, C., Wang, N., Chen, C.-Y., Choi, J., Agrawal, A., Shanbhag, N., & Gopalakrishnan, K. (2019). Accumulation Bit-Width Scaling For Ultra-Low Precision Training Of Deep Networks. ICLR. Retrieved from http://arxiv.org/abs/1901.06588

---

### Official Review · AnonReviewer3 · 2019-10-23
**Official Blind Review #3**

**Rating:** 3

**Review:**

In this paper, the authors propose a method to train models in FP16 precision. The authors show that the key reason of training performance drop is the overflow or underflow of back propagation information. Instead of using a fixed value or dynamic value proposed by a previous work, this paper adopts a more elaborate way to minimize underflow
in every layer simultaneously and automatically based on the current layer statistics. Experiment results on CIFAR10, ImageNet and Object Detection models are conducted to demonstrate the effectiveness of the proposed method.

There are some concerns about this paper:
1. This paper tries to solve a very practical problem which is good, however, the stability of this method which is very important for real applications remains unclear. More networks such as VGG/ResNet/depthwise-conv based networks, more initialization methods (such as gaussian, xavier, kaiming), w/o bn layers,  and more tasks such as segmentation and detection with different batch sizes are strongly recommended to make this work more solid.
2. In the experiments, it seems that dynamic loss scaling method works well too except on the configuration of SSD batchsize=8. Why dynamic loss scaling fails on this case? More detailed analysis are recommended to show the advantage of the proposed method.
3. In  many experiments, it seems adaptive loss scaling with FP16 is even better than FP32, is this stable? Could we further improve the FP32 results if using dynamic / adaptive loss scaling?

**Experience Assessment:**

I have read many papers in this area.

**Review Assessment: Checking Correctness Of Derivations And Theory:**

I assessed the sensibility of the derivations and theory.

**Review Assessment: Checking Correctness Of Experiments:**

I assessed the sensibility of the experiments.

**Review Assessment: Thoroughness In Paper Reading:**

I read the paper at least twice and used my best judgement in assessing the paper.

---

> ### Author Response · Authors · 2019-11-06
> **Responses for reviewer #3**
>
> We really appreciate your review.
>
> Q1:
> This is a great suggestion and we are working on to cover more DL workloads. But due to the limitation of time, we can only present the results on the networks mentioned in the paper, which are actually quite representative:
> 1. ResNet has been tested for image classification tasks on CIFAR and ImageNet;
> 2. VGG has been experimented since it is the backbone of our SSD example;
> 3. SSD does not use BN, which covers the w/o BN requirement;
> 4. We have also shown how different batch sizes may affect the SSD training result.
>
> Different initialization method is an interesting direction to try. But we think it may have a lower priority since we've shown that networks initialized by Kaiming, which is the initializer being widely used by default for training these networks, can be successfully trained by our approach.
>
> Q2:
> The failure scenario for dynamic loss scaling is basically due to its waste of training iterations. Dynamic loss scaling will not change the loss scale value unless it causes overflow. It is more like an automated trial-and-error method. Those iterations that cause overflow will be reverted. Therefore, if we have a fixed number of training iterations, dynamic loss scaling will waste quite a lot of them to trial loss scales that  cause overflow. In this way, the training will become less effective.
>
> See Fig 1. in https://github.com/ada-loss/ada-loss/issues/1
>
> This figure shows the frequency of dynamic loss scale values that cause overflow (each time loss scale reduces by half).
>
> Q3:
> Using FP16 can sometimes improve the test performance compared with FP32. This effect is more common for overparameterized models since they are more likely to overfit with FP32. FP16 can introduce some regularization error caused by less accurate arithmetic due to limited precision, and therefore, improve the test performance.
>
> Since FP32 can already cover the dynamic range of gradients in DNN training, it is not very necessary to use loss scaling for FP32 training in the first place. Applying other techniques may be more effective.
>
> It is worthwhile to mention that lower precision floating-point numbers can also benefit from adaptive loss scaling, like FP8 ([1]). We will explore this direction in the future.
>
> [1]. Mellempudi, N., Srinivasan, S., Das, D., & Kaul, B. (2019). Mixed Precision Training With 8-bit Floating Point. Retrieved from http://arxiv.org/abs/1905.12334

---

### Official Review · AnonReviewer1 · 2019-10-27
**Official Blind Review #1**

**Rating:** 6

**Review:**

The paper mostly reads well. It proposes to use statistics from previous activations to compute and adaptive scaling of the loss such that the amount of underflow is minimized. The scaling is defined per layer. Experiments are carried for various model sizes and datasets.

If anything I think the paper can do a better job at centralizing (maybe in an appendix) the gritty details (e.g. how the stats are computed etc). Unfortunately, the best way of doing this might be in the form of code or maybe pseudocode, but being quite explicit in all technical details.

Right now this is mentioned in the text (same way as batch norm stats if I understood correctly, based on the current minibatch). Though is not clear how the variance on w is treated. How is the variance on dirac delta (backpropagated error) is converted into a scalar (that will be for the entire loss).

**Experience Assessment:**

I do not know much about this area.

**Review Assessment: Checking Correctness Of Derivations And Theory:**

I did not assess the derivations or theory.

**Review Assessment: Checking Correctness Of Experiments:**

I assessed the sensibility of the experiments.

**Review Assessment: Thoroughness In Paper Reading:**

I made a quick assessment of this paper.

---

> ### Author Response · Authors · 2019-11-06
> **Responses for reviewer #1**
>
> Thank you so much for your kind comments.
>
> In general, stats for activation gradients are calculated in a similar way as batch norm, only during the backpropagation pass. For each layer (e.g., conv, fc), we will have two tensors, one is the gradient w.r.t. the mini-batch output activation, and the other is the weights tensor. The stats are collected by calculating the mean and variance on these two tensors. For more details, please refer to this function in our codebase - https://github.com/ada-loss/ada-loss/blob/060a6a8233c9b4af824bd2a9a46582d773545c0a/ada_loss/chainer_impl/ada_loss.py#L158-L176
>
> > How is the variance on dirac delta (backpropagated error) is converted into a scalar (that will be for the entire loss).
>
> We don't quite understand this question I'm afraid, would you mind adding more details about it? Thanks!

---

### Decision · Program_Chairs · 2019-12-19

**Decision:**

Reject

**Comment:**

This work proposes to improve mixed precision training by adaptively scaling the loss based on statistics from previous activations to minimize underflow during training. However, the method is designed rather heuristically and can be improved with stronger theoretical support and improved representation of the paper.